# Ataxin-2, Twenty-four, and Dicer-2 are components of a noncanonical cytoplasmic polyadenylation complex

Hima Priyanka Nadimpalli[1], Tanit Guitart[1,*], Olga Coll[1,*], Fátima Gebauer[1,2]

**Cytoplasmic polyadenylation is a mechanism to promote mRNA translation in a wide variety of biological contexts. A canonical complex centered around the conserved RNA-binding protein family CPEB has been shown to be responsible for this process. We have previously reported evidence for an alternative non-canonical, CPEB-independent complex in *Drosophila*, of which the RNA-interference factor Dicer-2 is a component. Here, we investigate Dicer-2 mRNA targets and protein cofactors in cytoplasmic polyadenylation. Using RIP-Seq analysis, we identify hundreds of potential Dicer-2 target transcripts, ~60% of which were previously found as targets of the cytoplasmic poly(A) polymerase Wispy, suggesting widespread roles of Dicer-2 in cytoplasmic polyadenylation. Large-scale immunoprecipitation revealed Ataxin-2 and Twenty-four among the high-confidence interactors of Dicer-2. Complex analyses indicated that both factors form an RNA-independent complex with Dicer-2 and mediate interactions of Dicer-2 with Wispy. Functional poly(A)-test analyses showed that Twenty-four and Ataxin-2 are required for cytoplasmic polyadenylation of a subset of Dicer-2 targets. Our results reveal components of a novel cytoplasmic polyadenylation complex that operates during *Drosophila* early embryogenesis.**

## Introduction

Changes in poly(A) tail length chiefly contribute to the stability and translation of mRNA (Passmore & Coller, 2021). Poly(A) tail elongation has been shown to promote translation in biological contexts as diverse as oocyte maturation, early embryonic development, neuronal plasticity, bone formation, cell proliferation, senescence, inflammation, metabolism, circadian gene expression, hibernation, and cancer (reviewed in Kojima et al [2012], D'Ambrogio et al [2014], Ivshina et al [2014], Grabek et al [2015], Kozlov et al [2021], Gewartowska et al [2021]). This process is typically controlled by the cytoplasmic polyadenylation element–binding (CPEB) family of proteins, which bind to U-rich cytoplasmic polyadenylation elements (CPEs) in the 3′ UTR of transcripts (Ivshina et al, 2014). Depending on sequence context and CPEB phosphorylation status, CPEBs can nucleate the formation of regulatory complexes that lead to either activation or repression of translation by balancing interactions of CPEB-associated factors with the mRNA cap and poly(A) tail (Piqué et al, 2008; Villalba et al, 2011; Fernández-Miranda & Mendez, 2012; Charlesworth et al, 2013; Ivshina et al, 2014). For example, phosphorylated CPEB1 recruits the cytoplasmic poly(A) polymerase Gld-2 to promote mRNA polyadenylation and translation, whereas unphosphorylated CPEB1 recruits silencing proteins that maintain the poly(A) tail short and the mRNA cap blocked.

CPEBs are highly conserved from humans to *Aplysia* (Ivshina et al, 2014). In *Drosophila*, the CPEB proteins Orb and Orb2 associate with the Gld-2 homolog Wispy to promote cytoplasmic polyadenylation (Cui et al, 2013; Norvell et al, 2015; reviewed in Kozlov et al [2021]). However, CPEB-independent cytoplasmic polyadenylation activities also exist in this organism, as well as poly(A) polymerases other than Wispy (Benoit et al, 2008; Coll et al, 2010; Dufourt et al, 2017). The RNAi-related endoribonuclease Dicer-2 promotes cytoplasmic polyadenylation and translation of *Toll* mRNA in early embryos (Wang et al, 2015; Coll et al, 2018). Dicer-2 associates with structured elements in the 3′ UTR of *Toll* and interacts with Wispy to promote polyadenylation (Coll et al, 2018). The extent of Dicer-2–mediated polyadenylation and the factors that cooperate with Dicer-2 in this endeavor have remained uncharacterized. Here, we perform RNA-immunoprecipitation (RIP) and Dicer-2 pull-downs to identify mRNA targets and cofactors of Dicer-2 in cytoplasmic polyadenylation. We find that Dicer-2 binds to a large number of mRNAs that are also bound by Wispy, suggesting pervasive roles in cytoplasmic polyadenylation. We identify 86 high-confident Dicer-2 protein interactors, most of which are unrelated to RNA interference, suggesting broader functions of Dicer-2 in cell biology. Of these, 21 are RNA-binding proteins, including the translation regulatory factors Twenty-four (Tyf) and Ataxin-2 (Atx2). We show that Tyf and Atx2 form an RNA-independent complex with

[1]Centre for Genomic Regulation (CRG), The Barcelona Institute of Science and Technology, Barcelona, Spain   [2]University of Pompeu Fabra (UPF), Barcelona, Spain

Correspondence: fatima.gebauer@crg.eu
Hima Priyanka Nadimpalli's present address is Center for Integrative Genomics (CIG), University of Lausanne, Lausanne, Switzerland.
*Tanit Guitart and Olga Coll contributed equally to this work.

Dicer-2 and Wispy, and that the Atx2/Tyf complex mediates the interactions of Dicer-2 with Wispy. Both Tyf and Atx2 are involved in cytoplasmic polyadenylation of a subset of Dicer-2 targets. These results reveal the composition of a noncanonical cytoplasmic polyadenylation complex and point to an emerging diversity of polyadenylation regulators beyond CPEB.

# Results and Discussion

### Identification of Dicer-2 mRNA targets

To identify the repertoire of RNAs that are bound by Dicer-2 in *Drosophila melanogaster*, we performed Dicer-2 immunoprecipitation (IP) from early (90 min) embryo extracts followed by RNA sequencing (RIP-Seq) in six biological replicates (see typical IP efficiency in Fig 1A). Parallel pull-downs with nonspecific IgG were carried as negative controls. RNAs significantly associated to Dicer-2 were determined by comparison with matched inputs and with control IgG IPs (Fig 1B and Table S1). We considered RNAs present in the Dicer-2 IP in an excess larger than 1.5-fold in both comparisons, and with a Benjamini-adjusted *P*-value < 0.001. This yielded a total of 1,749 RNAs associated to Dicer-2, the vast majority of which (>98%) were mRNAs (Fig 1C and D). These results agree with the observation that, despite the central role of Dicer in small RNA biogenesis, most Dicer-binding sites in human cells and *Caenorhabditis elegans* reside on mRNAs (Rybak-Wolf et al, 2014). These sites have been called "passive" because their recognition by Dicer does not result in mRNA cleavage. On average, Dicer-2 mRNA targets contain a higher GC content indicative of increased structure compared with non-targets (Fig 1E).

Transcripts known to be substrates of cytoplasmic poly-adenylation (*Toll*, *bicoid*) were associated to Dicer-2. To understand the potential of Dicer-2 as a cytoplasmic polyadenylation factor, we compared the list of Dicer-2 bound transcripts to that of Wispy polyadenylation substrates identified in previous studies (Cui et al, 2013; Eichhorn et al, 2016; Lim et al, 2016). One thousand eighty-two transcripts in the Dicer-2 list were also detected in at least one of the Wispy studies, indicating that a large fraction of Dicer-2 targets (62%) represent substrates of cytoplasmic polyadenylation (Fig 1F). Independent validation by RIP/RT-qPCR indeed confirmed binding of Dicer-2 to Wispy substrates, whereas abundant non-substrate mRNAs coding for ribosomal proteins were not bound (Fig 1G). These results suggest widespread roles of Dicer-2 in cytoplasmic polyadenylation.

### The Dicer-2 protein interactome

To identify the Dicer-2 protein interactome of early embryos, we performed co-immunoprecipitation (co-IP) using affinity purified αDicer-2 antibodies. As negative controls, we used either Dicer-2 null embryos or IP with nonspecific IgG, each comparison in biological triplicates. Because Dicer-2 null males are sterile, we obtained null embryo extracts by crossing homozygous Dicer-2 null females (*dicer-2^{L811fsX}*) with wild-type males (*w^{1118}*), whose contribution to the embryo protein content is only noticeable after 2 h of development once the maternal-to-zygotic transition has occurred (Fig 2A). Therefore, the male contribution to our early (90') embryo extracts is

negligible, as corroborated by Western blot (Fig 2B). We included treatment with RNase I to identify RNA-independent interactions. As expected, we observed co-immunoprecipitation of Dicer-2 with Wispy in an RNA-independent fashion and no background in the null or IgG controls (Fig 2B and C). Proteins present in the immunoprecipitates were then identified by tandem mass spectrometry (LC-MSMS).

Two different bioinformatic procedures were used to identify potentially true Dicer-2 interactors, SAINTexpress, and Top3 (Silva et al, 2006; Teo et al, 2014). SAINT (Significant Analysis of INTeractome) uses the spectral counts of all peptides in a protein, whereas Top3 uses the average peak areas of the three most intense peptides per protein. SAINT scored interactions for 854 proteins, of which 44 were found significant (Bayesian FDR ≤ 0.05) over the null control, whereas Top3 analysis revealed 123 significant interactors. Altogether, 128 proteins were found in the Dicer-2 IP over the null control, of which 39 were found using both SAINT and Top3 (Fig 2B, right and Table S2).

The analyses described above were also applied to find significant Dicer-2 interactors over the IgG control. We found a total of 635 interactors, of which 482 were both SAINT and Top3 hits (Fig 2C, right and Table S2). Overall, 86 proteins showed enrichment in the Dicer-2 IP against both the null and the IgG control and were categorized as the high-confidence Dicer-2 interactome (Fig 2D). Of these, 70 proteins (81%) were found to interact with Dicer-2 in an RNA-independent manner in all comparisons (Table S2). Therefore, the vast majority of interactions in the Dicer-2 high-confidence set are not mediated by RNA.

We next compared the Dicer-2 high confidence interactome with siRNA-associated factors previously identified by affinity chroma-tography against two different siRNAs in *Drosophila* embryo extracts (Gerbasi et al, 2010). Only two RNAi factors, R2D2 and Dicer-1, were found to overlap with our list, suggesting pervasive functions of Dicer-2 outside the RNAi machinery. Indeed, GO analysis indicated prevalence of cell division and biosynthetic processes, including translation regulation, among the terms most significantly associated with this group (Fig 2E).

### Ataxin-2 and Twenty-four mediate the interaction of Dicer-2 with Wispy

To narrow-down the partners of Dicer-2 potentially involved in cytoplasmic polyadenylation, we searched for Dicer-2–associated RNA-binding proteins (RBPs). We compared the Dicer-2 interactome with two recently published mRNA-bound proteomes of fly embryos (Sysoev et al, 2016; Wessels et al, 2016), revealing 21 RBPs as potential cofactors of Dicer-2 (Fig 3A and Table S2). Among these, we focused on RBPs involved in translational control. Notably, in addition to the translation regulators oskar, Fmr1, mushashi, Rasputin, and Wispy, we found two proteins, Twenty-four (Tyf) and Ataxin-2 (Atx2), previously shown to stimulate the synthesis of the clock protein Period in a manner that requires Poly(A)-binding protein (PABP) (Lim et al, 2011; Lim & Allada, 2013; Zhang et al, 2013). Furthermore, both the yeast Atx2 homolog Pbp1p and human ATXN2 have been shown to be involved in regulation of poly(A) tail length (Mangus et al, 1998, 2004; Inagaki et al, 2020). Thus, we hypothesized that Dicer-2, Atx2, Tyf, and Wispy may form an RNA-independent complex that works in cytoplasmic polyadenylation. Indeed, co-IP experiments indicated that these factors interacted in an RNA-independent fashion, suggesting that they formed a

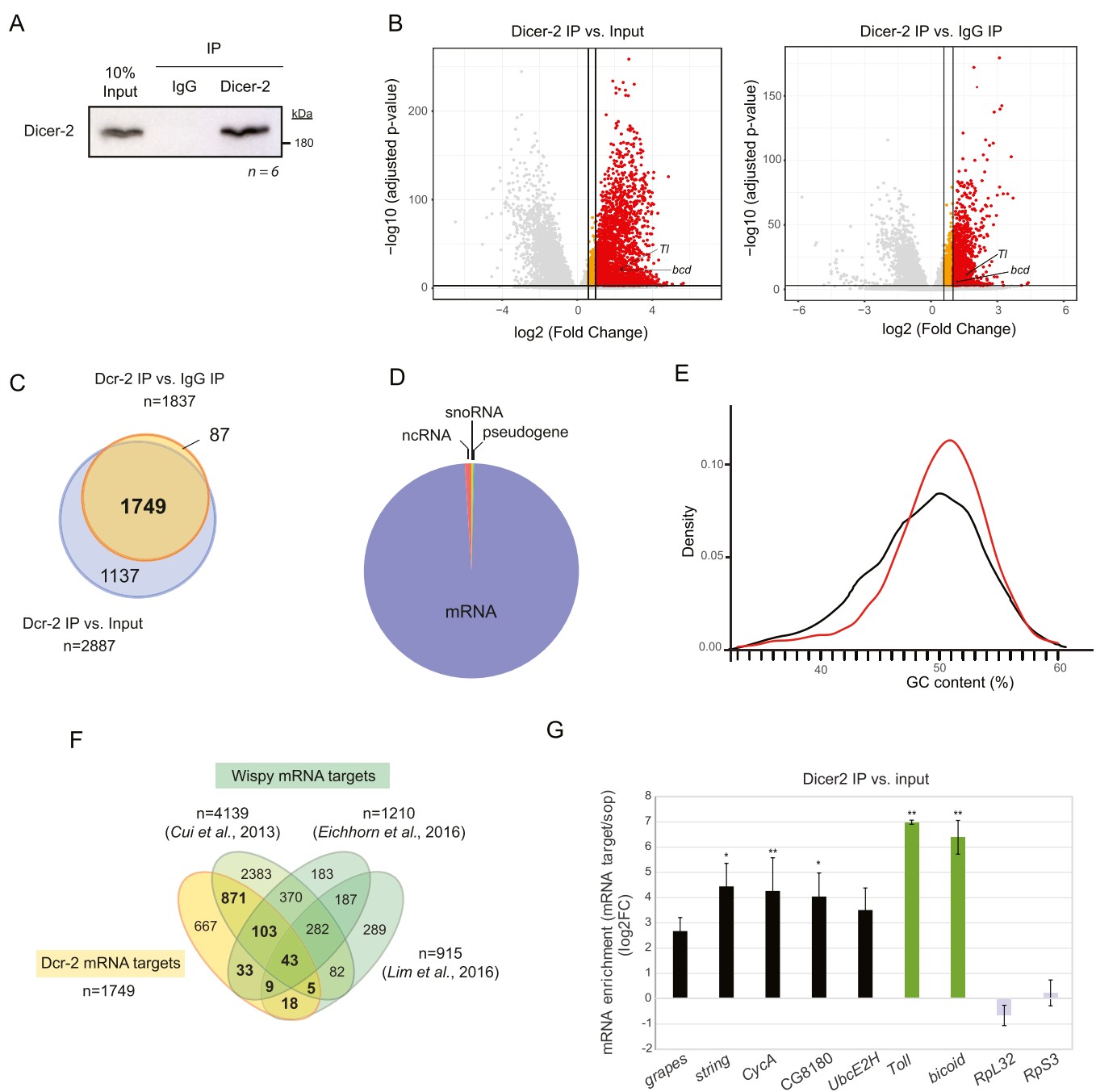

**Figure 1.  Identification of Dicer-2 mRNA targets.**
**(A)** Representative image of Dicer-2 immunoprecipitation. IgG, negative control. **(B)** Volcano plots showing RNA enrichment in Dicer-2 IPs with respect to input (*left*) or IgG (*right*). Red, fold enrichment > 2; orange, fold enrichment between 1.5 and 2; gray, not enriched. *Toll* (*Tl*) and *bicoid* (*bcd*) mRNAs are indicated. **(C)** Overlap between the two lists of potential Dicer-2 targets. **(D)** RNA types bound by Dicer-2. **(E)** GC-content of Dicer-2 targets (red) compared with non-targets (black). **(F)** Overlap of Dcr-2 and Wispy mRNA targets reported in three studies. Potential Dicer-2 polyadenylation targets are highlighted in bold. **(G)** Validation of RIP-seq by RT-qPCR. Selected targets of the Dcr-2/Wispy overlap were chosen. The enrichment was normalized to *sop* (*RpS2*), a ribosomal protein mRNA which does not undergo cytoplasmic polyadenylation. *Toll* and *bicoid* mRNAs (green) were used as positive controls, whereas *RpL32* and *RpS3* mRNAs (gray) were used as non-targets. Bar plots represent the average of three biological replicates. Error bars represent SD. Significance was assessed by *t* test (*P < 0.05, **P < 0.01).
Source data are available for this figure.

complex (Fig 3B). To corroborate complex formation and further understand mutual interactions, we performed co-immunoprecipitations in Tyf- or Atx2-depleted extracts. Depletions were performed by expressing UAS-RNAi constructs under the control of the maternal alphaTub67C driver, resulting in embryos with highly reduced Atx2 or Tyf levels (Fig 3C). Interestingly, depletion of Atx2 greatly affected the levels of Tyf and Wispy, whereas depletion of Tyf had a similar effect on Atx2 and Wispy levels, suggesting formation of a tightly

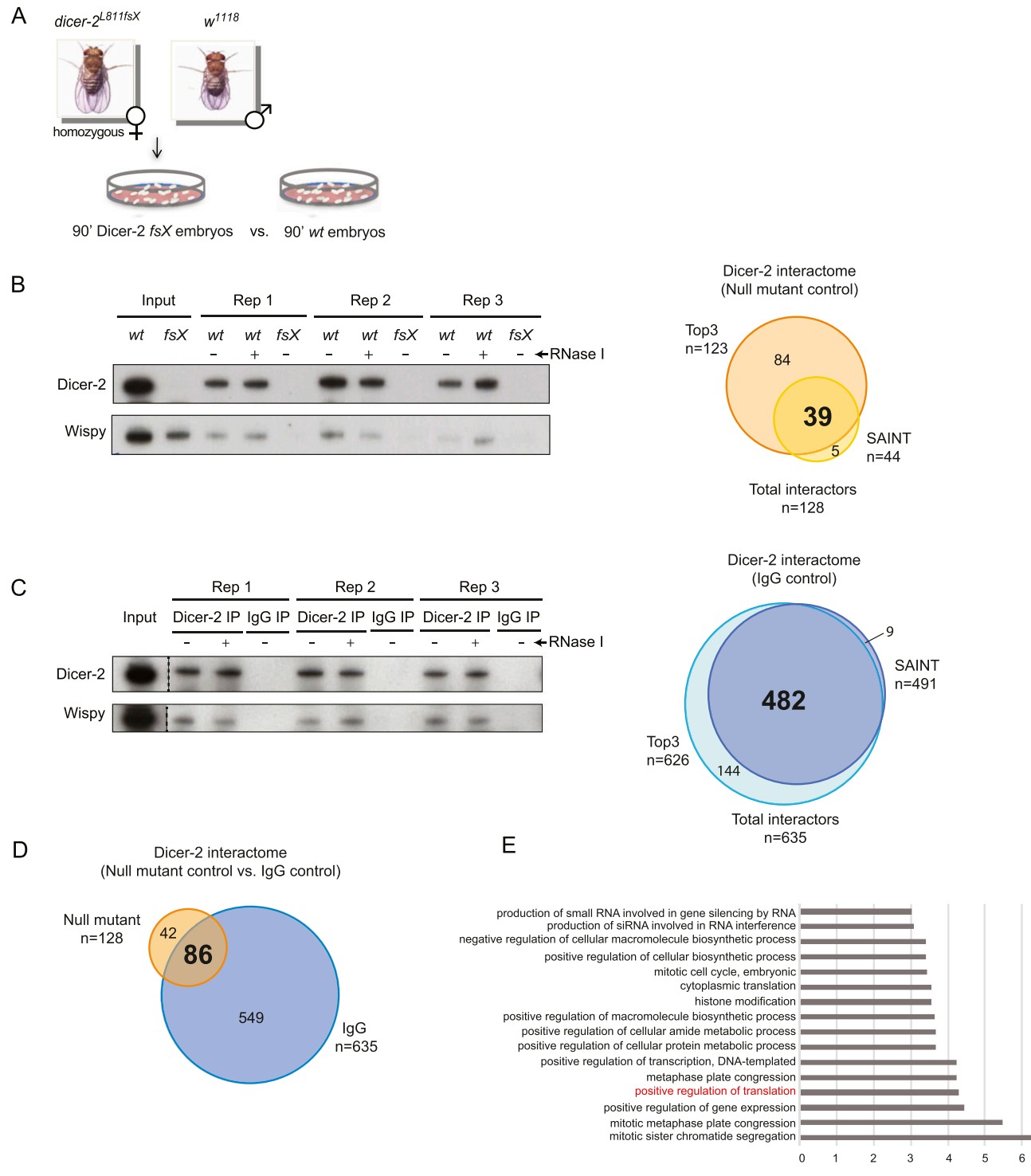

**Figure 2. Identification of the Dicer-2 protein interactome.**
**(A)** Schematic representation of the strategy to obtain Dicer-2 null embryos. **(B, C)** Comparison of wild-type ($w^{1118}$, *wt*) and Dicer-2 null (*fsX*) embryos (B) or αDicer-2 versus IgG IPs (C). In both cases, an immunoblot analysis of three replicates after treatment or not with RNase I is shown on the *left*, and the overlap between SAINT and Top3 analyses is shown on the *right*. **(D)** The Dicer-2 high confidence interactome, defined as the overlap between proteins identified in (B) and (C) (86 proteins). **(E)** Gene Ontology analysis of the Dicer-2 high-confidence interactome using Flyenrichr under the term "Biological Process."
Source data are available for this figure.

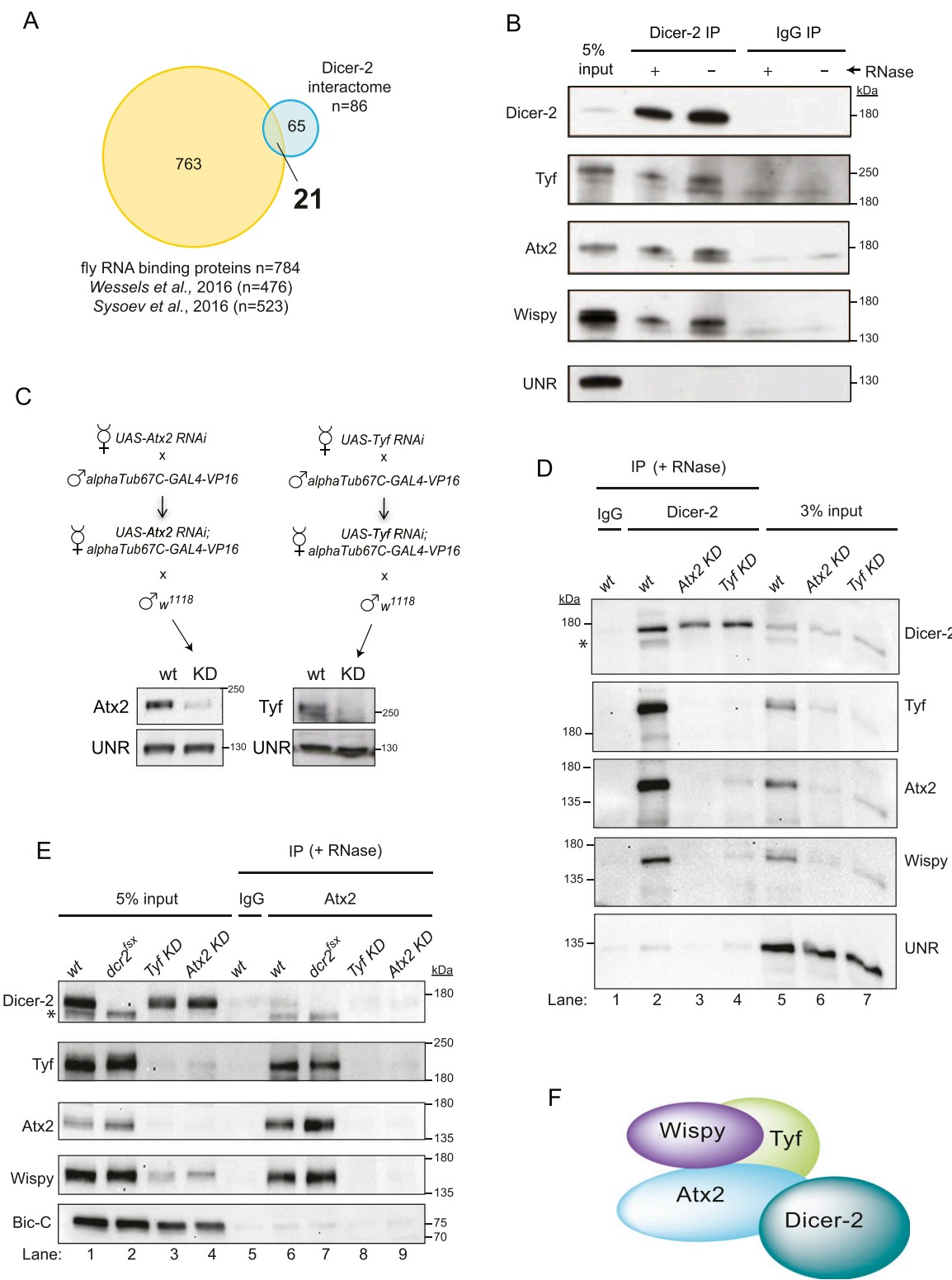

**Figure 3.   Dicer-2 interacts with Wispy through Ataxin-2 and Twenty-four.**
**(A)** Comparison between the Dicer-2 high-confidence interactome and the *Drosophila* RBPomes identified in 0–2 h embryos (Wessels et al, 2016) and across the maternal-to-zygotic transition (Sysoev et al, 2016). **(B)** Validation of Dicer-2 partners by co-immunoprecipitation. Dicer-2, Tyf, Atx2, and Wispy interact in an RNA-independent fashion. UNR was used as a negative control. **(C)** Schematic representation of the crosses performed to obtain Atx2- and Tyf-depleted extracts (*top*) and efficiency of depletion (*bottom*). **(D)** Interaction of Dicer-2 with Wispy is mediated by Atx2 and Tyf. Dicer-2 was immunoprecipitated from wild-type, Atx2-, and Tyf-depleted extracts in the presence of RNases. Co-IP of the indicated factors was assessed by Western-blot. IgG IP using wild-type extracts served as background control.

connected complex, as shown previously for splicing regulatory assemblies and other complexes (Eisenberg et al, 2018; Martín et al, 2021) (Fig 3D, lanes 5–7). Although Dicer-2 strongly co-precipitated with these factors, depletion of Atx2—and to a lesser extent Tyf—abrogated these interactions, indicating that the Atx2/Tyf complex mediates the interactions of Dicer-2 with Wispy (Fig 3D, lanes 2–4). To confirm these findings we pulled down Atx2 from wt, Dicer-2 null, Atx2-depleted and Tyf-depleted extracts. As in Fig 3D, the levels of Atx2, Tyf and Wispy were interdependent; however, they remained unaltered in Dicer-2 null extracts (Fig 3E, lanes 1–4). Atx2 interacted with Dicer-2, Tyf, and Wispy in wt extracts, corroborating complex formation (lanes 5–6). Atx2 interactions with Wispy and Tyf were fully preserved in Dicer-2 null extracts, indicating that they do not depend on Dicer-2 (compare lanes 6–7). As expected from very low levels of Atx2 in Atx2- or Tyf-depleted extracts, no IPs were detected in these conditions (lanes 8–9). Altogether, these results are consistent with a model in which Atx2/Tyf/Wispy form a complex that interacts with Dicer-2 (Fig 3F). Delineation of the full composition of this complex awaits further investigation.

### Twenty-four, Ataxin-2, and Dicer-2 work in cytoplasmic polyadenylation

The capacity of Dicer-2 to polyadenylate targets other than *Toll* mRNA was assessed using poly(A)-test (PAT) assays (Sallés & Strickland, 1995). We chose five targets across a range of *P*-values and found that their poly(A) tails were reduced in Dicer-2 null extracts (Fig 4A, left panels). Of these, three (grapes, CG8180 and string) were also targets of Atx2 and Tyf as their polyadenylation was reduced upon Atx2 or Tyf depletion (middle and right panels, see quantification of several experiments on the right). Not all targets, however, were dependent on Atx2 or Tyf, suggesting that alternative complexes may interact with Dicer-2 for cytoplasmic polyadenylation. The data, therefore, indicate that the Atx2/Tyf complex works in polyadenylation of a subset of Dicer-2 targets. Full identification of this subset will require high-throughput assessment of poly(A) tail changes upon alteration of Dcr-2, Atx2, and Tyf levels. Regardless, the results show that the functions of Atx2 in cytoplasmic polyadenylation are highly conserved and establish a new role for Tyf in this process, increasing the diversity of RBPs that functionally interact with cytoplasmic poly(A) polymerases (reviewed in Liudkovska and Dziembowski [2021]).

Atx2 has been shown to form mutually exclusive translation activation or repression complexes to regulate circadian clock periodicity and amplitude (Lee et al, 2017). The activation complex contains Lsm12 and Tyf, whereas the repression complex contains NOT1 and Me31B/DDX6, two proteins involved in deadenylation and miRNA-dependent silencing (Ostareck et al, 2014; Temme et al, 2014). Remarkably, Lsm12 is part of the Dicer-2 high confidence interactome, whereas NOT1 and Me31B are not (Table S2). These results support a model where Dicer-2 selectively interacts with the Atx2 activation complex to promote cytoplasmic polyadenylation and translation in early embryos (Fig 4B). Surprisingly PABP, an interactor of the Atx2/Tyf complex for translational regulation, was not included in the high-confidence Dicer-2 interactor list (Lim et al, 2011; Lim & Allada, 2013; Zhang et al, 2013). This is because PABP scored positive only in the Dcr-2 versus IgG IP comparison, suggesting that our strict criteria may exclude true interactors (Table S2). In sum, our results reveal the composition of a noncanonical cytoplasmic polyadenylation machinery that recruits Wispy for activation of maternal mRNAs, illustrating the growing diversity and plasticity of poly(A) tail length control assemblies.

# Materials and Methods

## Fly strains and generation of depleted and null embryos

*OregonR and w^{1118}* were used as wild-type flies. The following mutant or transgenic stocks were used: *w; Dcr2^{L811fsX}/CyO, amos^{Roi-1}* (kindly provided by Dr. Martine Simonelig), *w; UAS-Atx2-RNAi* (Vienna Drosophila Resource Center, VDRC #34955), *w; UAS-Tyf-RNAi* (VDRC #21965) and *w; alphaTub67C-GAL4-VP16; alphaTub67C-GAL4-VP16* (containing the driver in both chromosomes II and III, provided by Dr. Jerome Solon). Flies were maintained on standard food at 25°C.

*Dicer-2* null embryos were obtained by crossing *w; Dcr2^{L811fsX}* homozygous females with *w^{1118}* males. For the generation of Atx2-depleted embryos, homozygous virgin *w; UAS-Atx2-RNAi* females were crossed with homozygous *alphaTub67C-GAL4-VP16* males, and resulting females were then crossed with *w^{1118}* males. A similar strategy was followed to obtain Tyf depleted embryos using *w; UAS-Tyf-RNAi* females.

## Embryo extracts

For Dicer-2 interactome and RIP-Seq analysis, large-scale collections of staged *OregonR* 90 min embryos were obtained and processed as described in Coll et al (2010). For PAT assays, the protocol was scaled down as follows, carrying depleted/null and wild-type embryos in parallel. Egg-laying trays were exchanged three times for embryo synchronization, and 90 min embryos collected thereafter. In the case of Dcr-2 null embryos, because of their scarcity, multiple 90-min collections were taken in a period of 6–8 h, kept on ice, and pooled. Embryos were washed twice in EW buffer (0.7% NaCl, 0.04% TritonX-100), dechorionated in bleach (1:3 in EW buffer) for 2–3 min, and washed several times with milliQ water. Dechorionated embryos were then washed once with cold DEI buffer (10 mM Hepes, pH 7.4, 5 mM DTT, and 1× complete protease inhibitor from Roche) followed by homogenization in 1–2 embryo volumes of DEI buffer on ice. The homogenate was centrifuged at 10,000*g* for 10 min at 4°C. The intermediate cytosolic fraction was carefully collected, avoiding the upper lipid layer and

---

UNR was used as loading control. The asterisk denotes residual signal from Wispy detection. **(E)** Confirmation of complex formation and co-dependencies by Ataxin-2 pull-down. **(D)** Immunoprecipitation was performed as in (D). Bicaudal-C was used as loading control. **(F)** Model of the complex. Source data are available for this figure.

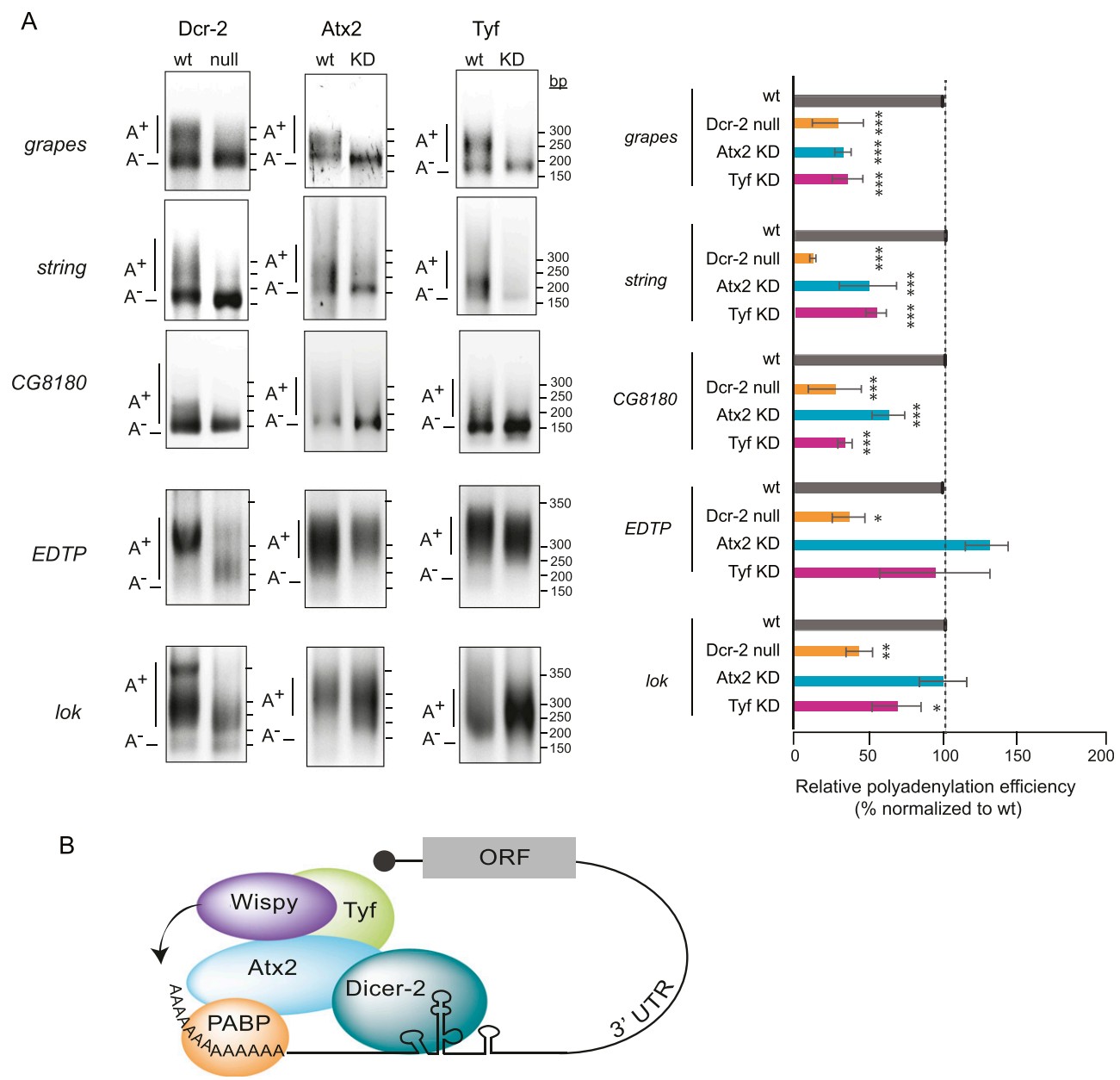

**Figure 4. Ataxin-2 and Twenty-four function in cytoplasmic polyadenylation of a subset of Dicer-2 targets.**
**(A)** Efficiency of polyadenylation of Dicer-2 targets in wild-type ($w^{1118}$) versus Dicer-2 null, Atx2-depleted, or Tyf-depleted embryos. The average of at least three independent experiments is shown on the right. Error bars represent SD. Significance was assessed using one-way ANOVA test with Bonferroni correction (*$P < 0.05$, **$P < 0.01$, ***$P < 0.001$). **(B)** Model for the embryonic noncanonical cytoplasmic polyadenylation complex. Interactions of Dicer-2 with Wispy, Atx2, and Tyf are depicted. Atx2 has been previously shown to interact with PABP.
Source data are available for this figure.

the pellet. Glycerol was added to 10%, and extracts aliquoted, snap frozen, and stored at –80°C.

## Antibodies

Rabbit αDicer-2 polyclonal antibodies were raised in house against the first 151 aa of the protein and affinity-purified using HiTrap NHS-activated HP columns (GE Healthcare) coupled with His-Dicer-2(1–257)-MBP (Coll et al, 2018). αWispy antibodies were kindly provided by Mariana Wolfner (Cui et al, 2008). αTyf and αAtx2 antibodies were provided by Joonho Choe (Lim et al, 2011) and Mani Ramaswamy (McCann et al, 2011), respectively. αBic-C antibodies were provided by Paul Lasko (Chicoine et al, 2007). αUNR antibodies have been described previously (Abaza et al, 2006). Antibody dilutions for Western blots were as follows: αDicer-2 (1:1,000), αTyf (1:500), αAtx2 (1:500), αWispy (1:2,000), αUNR (1:1,000), αBic-C (1:1,000).

## RNA immunoprecipitation followed by sequencing (RIP-Seq)

Protein-A Dynabeads (Invitrogen) containing affinity purified αDicer-2 antibody to their maximum binding capacity (2.4 μg antibody per 10 μl beads) were prepared by washing beads with 10 volumes of IPP 500 buffer (20 mM Hepes, 500 mM NaCl, 1.5 mM MgCl$_2$, 0.5 mM DTT, and 0.05% NP-40) followed by incubation with αDicer-2 antibody for 30 min at room temperature. Antibody-bound beads were washed twice with 10 vol of IPP 500 and IPP containing 150 mM NaCl (IPP 150). A similar procedure was followed to obtain beads with IgG used as negative control.

Embryo extracts were precleared with empty beads equilibrated in IPP 150 for 30 min at 4°C in the presence of 1× complete protease inhibitor cocktail (Roche). Precleared extracts (300–500 μg) were then added to 60 μl of antibody-bound beads in a final volume of 780 μl of IPP 150 and incubated at 4°C for 4 h on a rotating wheel. Following IP, beads were pelleted, washed four times with five volumes of IPP 150, and RNA extracted using TRIZOL. RNA quality was monitored in a Bioanalyzer. An aliquot of the IP was reserved for protein extraction and IP quality control.

Illumina True-seq libraries were prepared at the CRG Genomics facility using the Ribo-zero kit (Illumina) and subjected to 50-bp single-end sequencing. Reads were aligned to the *D. melanogaster* ENSEMBL genome release 87 (BDGP release 6_ISO1 MT/dm6 assembly) using the STAR mapper (version 2.5.2a) (Dobin et al, 2013). Qualimap (version 2.2.1) was used to check the quality of the mapped bam files (García-Alcalde et al, 2012). A raw count of reads per gene was obtained with HTSeq (htseq-count function, version 0.6.1p1) (Anders et al, 2015). The R/Bioconductor package DESeq2 was used to assess differential enrichment between experimental samples. Prior to DESeq2, genes for which the sum of raw counts across all samples was less than two were discarded.

## Dicer-2 protein interactome identification

Dicer-2 IPs were performed as indicated above, using 300 μg of wild-type (*w^1118*) or Dicer-2 null extracts, keeping the proportions of extract to beads and volumes. Following IP and washes, one half of the Dicer-2 beads were incubated with 100 U RNase I (Ambion) in 100 μl IPP 150 at 25°C for 15 min on a thermomixer. Untreated samples were incubated in parallel without addition of RNase I. Beads were then washed four times with five vol of IPP 150 before elution of proteins for mass spectrometry and Western blot. For mass spectrometry, samples on beads were washed thrice with 500 μl of 200 mM ammonium bicarbonate (ABC) and resuspended in 60 μl of 6 M urea in 200 mM ABC. Proteins were then reduced by adding DTT (10 μl DTT 10 mM, 37°C, 60 min), alkylated with iodoacetamide (10 μl of IAM 20 mM, 25°C, 30 min), diluted with 200 mM ABC to reach a final urea concentration of 1M, and digested overnight with trypsin (1 μg, 37°C). The peptide mixture was collected and acidified to a final concentration of 5% formic acid. Samples were desalted using a C18 column, evaporated to dryness, and diluted to 10 μl with 0.1% formic acid in milliQ water. Forty-five percent of the peptide mixture was analyzed by LC-MSMS using a 1-h gradient in the LTQ-Orbitrap Velos Pro mass spectrometer (Thermo Fisher Scientific). The data were acquired with Xcalibur software v2.2. and analyzed using the Proteome Discoverer software suite (v1.4.1.14; Thermo

Fisher Scientific). The search engine Mascot (v2.5.1 Matrix Science; Perkins et al, 1999) was used for peptide identification. Peptide data were searched against the Uniprot (UP_Drosophila) protein database, and the identified peptides were filtered to a threshold of 5% FDR.

Protein–protein interactions were assessed using SAINTexpress (Teo et al, 2014) and Top3 (Silva et al, 2006) analysis. SAINT interactors with Bayesian false discovery rate of <0.05 were included in the Dicer-2 interactome. For Top3 analysis, the log$_2$ of the average area of the three most intense peptides of each protein was calculated by Proteome Discoverer. Top3 analysis was constrained by very low abundance or the absence of several proteins in the null IP and by missing peak area values for one of three null mutant replicates. These technical caveats were overcome by semiquantitative scoring of significant peptide occurrences in wild-type versus null replicates and by imputation that substitutes the missing values for the mean value of other replicates, respectively. A *t* test was performed between the three replicates of each sample condition to identify differentially abundant proteins.

## Small scale immunoprecipitation

Protein-A dynabeads (20 μl) were equilibrated in wash buffer (10 mM Hepes pH 8.0, 8% Glycerol) and blocked by incubation with 700 μg embryo extract for 1 h at room temperature. After extensive washing, 2 μl Atx2 serum, 5 μg of affinity-purified αDicer-2 antibody, or IgG were added in wash buffer followed by incubation for 1 h at room temperature and three additional washes. Beads were mixed with 700 μg of embryo extract in 200 μl DE buffer (10 mM Hepes, 5 mM DTT), supplemented with 50 μl of buffer containing 20 mM Hepes, pH 8, 150 mM NaCl, 0.1% NP-40, 1 mM EDTA, 0.5 mM DTT, and 1× protein inhibitor cocktail (Roche) and incubated at 25°C for 1 h on a rotating wheel. Beads were then washed three times with five volumes of 1× NET (50 mM Tris–HCl pH7.5, 150 mM NaCl, 0.1% NP-40, 1 mM EDTA). RNase treatment was performed with 100 U RNase I (Ambion) and 10 μg of RNase A (Sigma-Aldrich) in 40 μl wash buffer with protein inhibitors at room temperature for 20 min. Untreated samples were incubated in parallel without adding RNases. Beads were washed three times with 10 volumes of 1xNET and proteins eluted in SDS buffer for Western blot.

## PAT assays

Pools of 100–200 embryos synchronized at 60–90 min were collected in Eppendorf tubes and disrupted with Pellet mixer Cordless (Avantor) in 200 μl Trizol. RNA was obtained using a Maxwell 16 LEV simplyRNA kit (Promega). One microgram of total RNA was used for PAT assays as described in Sallés and Strickland (1995). In some experiments, the poly(A) tail assay kit from Thermo Fisher Scientific (Ref. 764551KT) was used with equivalent results. Gene-specific forward oligonucleotides used for PAT assays were as follows: String (5′-CCGAAACGCAAATGCAAAC-3′), Grapes (5′-CATTGCATTGTTTACGAGTACG-3′), CG8180 (5′-CAGCAGCATTAACTGACGAATCGAC-3′), EDTP (5′-GGCT AAACCGCTCTCCTGTTCTCTAG-3′) and lok (5′-CTGCATTTAAACTGGGCTG CTGCTTC-3′). Amplified products were resolved in agarose gels, and the efficiency of polyadenylation was measured as the ratio of polyadenylated (A+) versus non polyadenylated (A−) RNA, quantified using

ImageQuant-TL. The data for Atx2-depleted, Tyf-depleted and Dicer-2 null embryos were normalized to the wt control (set to 100%) that was carried in parallel in each experiment.

## RT-qPCR

RNA was reverse-transcribed with SuperScript II (Invitrogen) using a mix of random primers and oligo(dT). The cDNAs were subjected to quantitative real-time PCR using SYBR Green PCR master mix (Applied Biosystems) and ViiA 7 Real-Time PCR System (Applied Biosystems) following the manufacturer's instructions. Oligonucleotides used in qPCR are as follows: Toll-f (5'-CGCTGCCTT CGCGTCTGTTTGC-3'), Toll-r (5'-GTGTGGAATGCTCGAATAAGTCA-3'), Bicoid-f (5'-ATTGCAATCTGTTAGGCCTCAAG-3'), Bicoid-r (5'-CGGGAT CCCGAGTAGAGTAGTTCTTATATATT-3'), Sop-f (5'-CCGTGGTACTGGCAT TGTCT-3'), Sop-r (5'-CCGAGTATGCCTGGTAAGGA-3'), Grapes-f (5'- TGAGGAGAATGACCCGATTC-3'), Grapes-r (5'-AACCACCCAGTCTTTCGAT G-3'), String-f (5'-CACAAGCGCAACATCATTATC-3'), String-r (5'-CCGG ATAGGCGTTGGTATT-3'), CycA-f (5'-GCTGGAGGAGATCACGACTT-3'), CycA-r (5'-TTGTACTTTTCCCGCATGG-3'), CG8180-f (5'-AAGGTCCCAA- CATCTCTCTGA-3'), CG8180-r (5'-GCTGGGGATGGTATCTCGTA-3'), UbcE2H-f (5'-CAACGATCGCAACAGATCAT-3'), UbcE2H-r (5'-TTTTGCTCTTCCATCC GTTC-3'), RpL32-f (5'-TGCCCACCGGATTCAAGA-3'), RpL32-r (5'-AAAC GCGGTTCTGCATGAG-3'), RpS3-f (5'-CGATTTCCAAGAAACGCAAG-3'), RpS3-r (5'-CGAGTCAGGAACTCGTTCAA-3').

# Data Availability

The RNA-Seq datasets produced in this study are available in Gene Expression Omnibus with identifier GSE189868. The mass spectrometry proteomics data have been deposited to the ProteomeXchange Consortium via the PRIDE partner repository with the dataset identifier PXD030281.

# Supplementary Information

# Acknowledgements

We thank Mariana Wolfner, Paul Lasko, Joonho Choe, and Mani Ramaswamy for kindly sharing their antibodies, and Martine Simonelig and Jerome Solon for fly strains. We are grateful to Eva Novoa, Rebeca Medina, and Ana Milovanovic for useful suggestions and data not shown. We also thank the CRG Proteomics, Genomics, Bioinformatics, and Protein Technologies Facilities for protein identification, RNA sequencing, data analysis, and αDicer-2 affinity purification, respectively. H-P Nadimpalli was supported by a La Caixa fellowship. This work was supported by grants from the Spanish Ministry of Science and Innovation (PGC2018-099697-B-I00 and BFU2015-68741) and the Catalan Agency for Research and Universities (2017SGR534). We acknowledge the support of the Spanish Ministry of Science and Innovation through the Centro de Excelencia Severo Ochoa (CEX2020-001049-S, MCIN/AEI/10.13039/501100011033) and the Generalitat de Catalunya through the CERCA Programme.

## Author Contributions

HP Nadimpalli: conceptualization, data curation, formal analysis, validation, investigation, methodology, and writing—review and editing.
T Guitart: data curation, formal analysis, validation, investigation, and writing—review and editing.
O Coll: validation, investigation, methodology, and writing—review and editing.
F Gebauer: conceptualization, supervision, funding acquisition, project administration, and writing—original draft, review, and editing.

## Conflict of Interest Statement

The authors declare that they have no conflict of interest.

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
