## [Reviewer comments · Life Science Alliance]

Life Science Alliance

Atx2, Tyf and Dcr-2 are components of a non-canonical cytoplasmic polyadenylation complex

Hima Nadimpalli, Tanit Guitart, Olga Coll, and Fatima Gebauer

DOI: 10.26508/lsa.202201417

Corresponding author(s): Fatima Gebauer, Centre for Genomic Regulation

Review Timeline:

Submission Date:	2022-02-18
Editorial Decision:	2022-02-24
Revision Received:	2022-07-19
Editorial Decision:	2022-08-08
Revision Received:	2022-08-22
Editorial Decision:	2022-08-22
Revision Received:	2022-09-05
Accepted:	2022-09-06

Transaction Report:

Please note that the manuscript was reviewed at Review Commons and these reports were taken into account in the decision-making process at Life Science Alliance.

February 24, 2022

Re: Life Science Alliance manuscript #LSA-2022-01417

Prof. Fatima Gebauer
The Barcelona Institute of Science and Technology

Dear Dr. Gebauer,

Thank you for submitting your manuscript entitled "Ataxin-2, Twenty-four and Dicer-2 are components of a non-canonical cytoplasmic polyadenylation complex" to Life Science Alliance. We invite you to re-submit the manuscript, revised according to your Revision Plan.

Thank you for this interesting contribution to Life Science Alliance. We are looking forward to receiving your revised manuscript.

Sincerely,

- A letter addressing the reviewers' comments point by point.
- An editable version of the final text (.DOC or .DOCX) is needed for copyediting (no PDFs).
- High-resolution figure, supplementary figure and video files uploaded as individual files: See our detailed guidelines for preparing your production-ready images, <https://www.life-science-alliance.org/authors>
- Summary blurb (enter in submission system): A short text summarizing in a single sentence the study (max. 200 characters including spaces). This text is used in conjunction with the titles of papers, hence should be informative and complementary to the title and running title. It should describe the context and significance of the findings for a general readership; it should be written in the present tense and refer to the work in the third person. Author names should not be mentioned.
- By submitting a revision, you attest that you are aware of our payment policies found here: <https://www.life-science-alliance.org/copyright-license-fee>

B. MANUSCRIPT ORGANIZATION AND FORMATTING:

*****IMPORTANT:** It is Life Science Alliance policy that if requested, original data images must be made available. Failure to provide original images upon request will result in unavoidable delays in publication. Please ensure that you have access to all original microscopy and blot data images before submitting your revision. *******

Rebuttal letter:

Manuscript #LSA-2022-01417

We thank the reviewers for their useful suggestions. We have improved our manuscript by performing new experiments addressed at characterizing the Dicer-2/Atx2/Tyf/Wispy complex, resulting in a full new Figure (Figure 3), and have provided additional Dicer-2 polyadenylation targets (Figure 4A). Some of the proposed experiments have inherent technical difficulties that we have been so far unable to resolve (please, find detailed explanations below). Despite these difficulties, we believe we provide solid evidence for a non-canonical polyadenylation complex operating in *Drosophila* embryos, and hope that the reviewers find our manuscript acceptable for publication.

Reviewer #1

This is a potentially interesting study that aims at identifying substrates of Dicer-2 regulated Orb1 mediated cytoplasmic polyadenylation in *Drosophila* embryos. The authors used RIP-seq to identify interacting RNAs and Ip MS to look for proteins interactions of DICER-2. They propose that Atx2 and Tyf proteins cooperate with DICER-2 and Orb1 which is supported by silencing and PAT assays.

We thank the reviewer for considering our work potentially interesting. We would like to clarify, however, that we do not investigate the CPEB homolog Orb1 because it is not expressed in Drosophila embryos, where Orb-2 takes the lead. We believe the reviewer may have meant 'Wispy' rather than 'Orb1'. We precisely argue that the cytoplasmic polyadenylation complex we describe is 'non-canonical' because it centres around Dicer-2, a factor we have previously shown to drive cytoplasmic polyadenylation in an Orb2-independent fashion (Coll et al., RNA 2018; Coll et al., Genes Dev 2010).

Major concerns:

1) RNA-protein interactions are often transient therefore RIP-seq assay which allows for multiple associations and dissociations is not ideal for deciphering interaction patterns in vivo. Especially that the Authors considered 1.5-fold change as an indication for a significant interactor. I would suggest much higher stringency. Moreover, I would suggest validating the data using a cross-linking based methodology, such as CLIP which is much more reliable.

*We agree with the reviewer that using CLIP would provide a sound description of Dicer-2 targets in vivo. However, this technology depends on efficient UV-crosslinking of proteins to RNA, and is notoriously difficult to perform in Drosophila embryos, where the egg shell precludes UV-light from reaching the deepest volume of the embryo (e.g. see Sysoev et al., Nat Commun 2016). Furthermore, Dicer-2 binds to double-stranded RNA and does not crosslink efficiently to targets using the 254 nm wave length of CLIP analysis. In our hands, even recombinant Dicer-2 was unable to crosslink to RNA in vitro. As an alternative, we tried to use PAR-CLIP, which has been previously used to identify DICER targets in mammalian cells and *C. elegans* (Rybak-Wolf et al., Cell 2014). PAR-CLIP confers increased UV-crosslinking efficiency at 365 nm compared to conventional UV-crosslinking at 254 nm. PAR-CLIP in flies requires the generation of animals expressing the UPRT gene, which converts 4-thiouracil (4TU, provided to flies in the food) to 4-thiouridine (4SU) that can be incorporated into nascent RNA (Wessels et al., Genome Res 2016). For PAR-CLIP in embryos, we generated*

flies expressing the UPRT gene under the nanos promoter, and collected 90 min embryos laid by UPRT females. RNAs, however, showed low 4SU incorporation efficiency (Figure R1) which proved insufficient to obtain reliable PAR-CLIP libraries.

Figure R1. Assessment of 4SU incorporation in UPRT-expressing flies. A) A weak level of 4SU incorporation is detected in embryos from UPRT-expressing mothers. Flies were fed with increasing amounts of 4TU, and RNA from embryos was isolated. The thiol group of 4SU was biotinylated using HPDP-biotin conjugate, and visualized by streptavidin hybridization in a Dot-blot assay. RNA from embryos obtained in the absence 4TU (0) served as negative control. In vitro biotinylated RNA served as positive control. B) 4SU-incorporation in UPRT-expressing flies is slightly over the background detected in *wt* flies fed with 4TU.

We thereby decided to use RIP-Seq. RIP-Seq has been extensively used to identify in vivo mRNA targets of RNA-binding proteins, including targets of Atx2 in Drosophila brains (Rounds et al., Genetics 2022). Although RIP-Seq may result in spurious interactions due to homogenization of the sample, our samples are in themselves very homogeneous, as they are staged 90 minute embryos consisting of a syncytium (i.e. no different tissues present, just single syncytial cells). In addition, contrary to CLIP that detects even very transient interactions, RIP-Seq detects stable interactions (reviewed in Wheeler et al., WIREs RNA 2017). Thus, RIP-Seq can be used to identify reliable protein-RNA interactions in vivo.

Regarding the thresholds that we use, in Figure 1B we show that the difference in the number of targets between considering 1.5 (orange) and 2-fold (red) changes is minimal and, thus, we decided for the 1.5 fold-change in order to be more inclusive. A 2-fold change threshold is common in RIP-Seq analysis and, in our case, preserves identification of known Dicer-2 targets such as Toll and bicoid (indicated in Figure 1B). Using a higher threshold would result in loss of these known targets, indicating that many true targets of Dicer-2 could also be lost. In our view, rather than thresholds which are by definition arbitrary, it is important that significant targets are identified no matter the type of analysis performed. In this sense, we have considered targets identified in both the 'Dicer-2 IP vs input' and the 'Dicer-2 IP vs IgG' comparisons as the list of reliable Dicer-2 targets.

2) The PAT assays presented by the Authors, are not very convincing. I would suggest applying either PAT-seq or preferably a genome-wide approach (TAIL-seq, PAL-seq or Nanopore sequencing).

We agree with the reviewer that high-throughput assessment of poly(A)-tail changes would be optimal. In the past, we tried TAIL-Seq with the advice of Narry Kim when she first described this technology. The difficulty was that Dicer-2 null flies are sterile and females lay very few eggs. Thus, we spent about a year collecting RNA from Dicer-2 null embryos in order to obtain sufficient material to perform Tail-Seq, only to find out that just one library out of triplicates showed the expected dynamic range in detection of poly(A) tail lengths of the spike-in controls. Thus, Tail-Seq was unsuccessful for us.

To address the reviewer concern we tried nanopore sequencing in collaboration with the expert lab of Eva Novoa (CRG-Barcelona). During the time-frame of the revision we could collect sufficient amount of RNA to perform duplicates. The results unfortunately showed strong batch effects which precluded identification of poly(A) tail changes with statistical significance. Batch effects are expected, as our samples are embryos pulled in different collections from different mothers. Because obtaining reliable results for nanopore analysis would require extensive efforts of sample collection that would significantly extend the time of revision of this manuscript, we opted for providing more convincing PAT images. To this end, we compared our classical PAT method with the state-of-the-art PAT kit from Thermofisher, but could not find major improvements with the kit (see Figure R2).

Figure R2. PAT assay comparison. The polyadenylation status of *grapes* mRNA was determined using either our classical PAT method or the poly(A) tail assay kit from Thermofisher (Ref. 764551KT) in wt vs *dcr2* null (left panel), wt vs *Atx2* KD (middle panel) and wt vs *Tyf* KD (right panel) embryo extracts. M, 50 bp ladder. For the middle and right panels, images of the classical method have been taken from the paper. The same extracts were used to perform PAT assays with the kit.

We therefore finally decided to improve Figure 4A by, first, increasing the contrast of existing panels and, second, substituting the non-target *CycA* and the weak target *UbcE2H* with two new targets. The data are shown in Figure 4A and discussed in page 7 of the manuscript, where we acknowledge that additional high-throughput experiments are required: 'The data, therefore, indicate that the *Atx2*/*Tyf* complex works in polyadenylation of a subset of *Dicer-2* targets. Full identification of this subset will require high-throughput assessment of poly(A) tail changes upon alteration of *Dcr-2*, *Atx2* and *Tyf* levels.'

Reviewer #2

In this short report, Nadimpalli et al. investigate a role for *Drosophila* *Dicer-2* in cytoplasmic polyadenylation. *Dicer-2* has been characterized primarily for its role in siRNA processing and RISC loading. However, work by Wang et al. (2015, *Sci. Adv.*) -notably not referenced here - showed that *Dicer-2* binds to the 3'UTR of *Toll* mRNA in vitro and in vivo and regulates *Toll*

translation, suggesting an siRNA-independent function of *Dicer-2* in translational activation. Subsequently, the Gebauer lab showed that *Dicer-2* interacts with the cytoplasmic poly(A) polymerase *Wispy* and that both *Dicer-2* and *Wispy* are required for cytoplasmic polyadenylation and translational activation of *Toll* in the *Drosophila* embryo.

We apologize for this oversight. The paper by Wang et al is now mentioned in the manuscript.

In the current manuscript, the authors use co-immunoprecipitation methods to identify 1) other proteins involved in Dicer-2/Wispy-mediated polyadenylation and 2) the spectrum of Dicer-associated RNA targets. This study goes beyond the previous work by providing evidence suggesting that Dicer-2 is broadly involved in cytoplasmic polyadenylation (i.e. numerous targets identified), but falls short because the authors do not test the ultimate relevance of this function - whether there is an effect on translation of the target RNAs. Among proteins identified as Dicer-2 interactors, the authors show that Atx2 and Tyf also play a role in cytoplasmic polyadenylation. Human Atx2 was previously shown to regulate cytoplasmic polyadenylation so the current work indicates that the role is conserved. Perhaps

more interestingly, Tyf was previously shown to activate translation of circadian clock mRNAs and this study suggests that it may act more broadly in translational activation through cytoplasmic polyadenylation. Here, too, testing the connection between cytoplasmic polyadenylation and translation would greatly strengthen the manuscript.

We agree with the reviewer that the connection with translation would provide a valuable layer of information to the manuscript. However, we think that this information is not essential to support our main conclusion that Atx2, Tyf and Dicer-2 are components of a non-canonical cytoplasmic polyadenylation complex. Whatever the consequences that cytoplasmic polyadenylation may have on translation, we provide clear evidence that Atx2 and Tyf are involved in polyadenylation of Dicer-2 targets, and that these factors interact with the cytoplasmic poly(A) polymerase Wispy.

Major points

1) The conclusion "Therefore, the vast majority of interactions in the Dicer-2 high confidence set are direct" is based on the RNA independence of these interaction. RNA-independence is not a measure of directness, since proteins with multiple degrees of separation may be immunoprecipitated together (e.g., proteins in a complex). Thus, this conclusion needs to be tempered.

We have tempered this conclusion by eliminating the word 'direct' from the sentence, which now reads as follows: 'Therefore, the vast majority of interactions in the Dicer-2 high-confidence set are not mediated by RNA'.

2) The model put forth is that Dicer-2, Atx-2, Tyf, and Wispy form a complex that promotes cytoplasmic polyadenylation. However, no evidence is provided to corroborate the presence of a complex of Dicer-2, Atx-2, Tyf, and Wispy as the authors state: "Indeed, co-IP experiments corroborated the presence of such a complex (Figure 2G)." The only co-IP experiments presented show that Dicer-2 can interact with each of these proteins, but there are no data showing that they interact with each other (or are in a complex with each other) - there could be multiple independent complexes with Dicer-2. It is essential to perform a series of co-IPs using antibodies or tags for each protein of the group. In addition, co-IPs from RNAi embryos/mutants would be informative (e.g., does recruitment of Wispy to Dicer-2 require Atx2 or Tyf?; does Wispy interact with Atx2 or Tyf independently of Dicer-2?, does Dicer-2 interact with Atx2 independently of Tyf and vice versa). And lastly, it is odd that

the authors don't provide more evidence for their model (Figure 3C) by testing whether PABP is in their IPs and using RNAi/mutants to test the connectivity (e.g., does interaction of Dicer-2 with PABP require Atx2?).

As the reviewer suggests in the previous point, finding RNA-independent interactions is suggestive of 'proteins in a complex'. We agree that additional co-IP experiments would strengthen this conclusion, especially if these IPs are performed with the endogenous proteins. We therefore performed additional co-IPs with Atx2 antibodies in Dicer-2 null, Tyf-KD and Atx2-KD extracts, and with Dicer-2 antibodies in Atx2-KD and Tyf-KD extracts, in the presence of RNases. The results confirm the existence of the Dicer-2/Atx2/Tyf/Wispy complex, and reveal that Atx2/Tyf mediate the binding of Wispy to Dicer-2. These data have been included in new Figure 3.

Regarding PABP, we have included it in the model because Atx2 has been previously shown to contact PABP in several organisms (Lim and Allada, Science 2013; Zhang et al., Science 2013; Mangus et al., MCB 1998; Inagaki et al., JBC 2020). We do not find PABP as an interactor of Dicer-2 (hence, it is not found in Table 2) and we hope to have made this clear in the model by depicting PABP binding to Atx2 but not to Dicer-2. We have now indicated this distinction in the figure legend.

3) How were the PAT assay gels actually quantified to obtain the values in the graphs Figure 3B? This needs to be described in the Methods. Also, are the wild-type values derived from the control used for particular one of the three target tested or did the authors average wild-type from 3 experiments for each target and then take the average of these? The latter is not a correct way to present the data. Lastly, the T-test is not the correct statistical test - an ANOVA with a post-hoc test should be used when there are multiple comparisons to the control (avoid type I error). Lastly, what are the y-axis values in the graphs?

We thank the reviewer for spotting this oversight. The Y-axis is now properly labelled and the description of quantification has been included in Methods as follows:

'Amplified products were resolved in agarose gels, and the efficiency of polyadenylation was measured as the ratio of polyadenylated (A+) versus non polyadenylated (A-) RNA, quantified using ImageQuant-TL. The data for Atx2-depleted, Tyf-depleted and Dicer-2 null embryos were normalized to the wt control (set to 100%) that was carried in parallel in each experiment'. Thus, each wt-test comparison is taken as a separate datapoint (that is, we do not average all wild types), and the average of those is represented in the graphic together with the standard deviation. We have now performed one-way ANOVA test with Bonferroni correction using GraphPad Prism, and have modified the Figure accordingly.

4) Regarding the conclusion "These results also show that functions of Atx2 in cytoplasmic polyadenylation are highly conserved, and establish a new role for Tyf in this process, increasing the diversity of RBPs that functionally interact with cytoplasmic poly(A) polymerases", what do the authors mean by functionally interact? There are no data presented showing that Tyf or Atx2 interact with Wispy, only that they interact with Dicer-2 and no genetic interaction data either.

By 'functional interaction' we mean RBPs that are necessary for cytoplasmic polyadenylation, as shown in the PAT assays of Figure 4A. In addition, we provide now additional data confirming the Dicer-2/Atx2/Tyf/Wispy complex.

5) How do the PAT assay results indicate that "Atx2, Tyf and Dicer-2 act in concert for cytoplasmic polyadenylation of Dicer-2 targets"? The data show a requirement for each individually, but nothing

This comment seems to have been cut during submission and is not complete. If we understand properly, the reviewer may be referring to the fact that he/she is not convinced that Atx2, Tyf and Dicer-2 form a complex, something that we have strengthened with the new co-IPs shown in Figure 3. We think the proteins act in concert because they are part of a complex, and because Atx2 and Tyf depletion reduces polyadenylation of a subset of Dicer-2 targets. Nevertheless, we have eliminated this sentence from the manuscript.

Minor points

1) How was the 1.5-fold threshold for Dicer-2 associated RNAs set? It seems like very low threshold to use.

This comment was also raised by reviewer #1. In Figure 1B we show that the difference in the number of targets between considering 1.5 (orange) and 2-fold (red) thresholds is minimal and, thus, we decided for the 1.5 fold-change in order to be more inclusive. A 2-fold threshold is common in RIP-Seq analysis and, in our case, preserves identification of known Dicer-2 targets such as Toll and bicoid (indicated in Figure 1B). Using a higher threshold would result in loss of these known targets, indicating that other true targets of Dicer-2 could also be lost. In our view, rather than thresholds which are by definition arbitrary, it is important that significant targets are identified no matter the type of analysis performed. In this sense, we have considered targets identified in both the 'Dicer-2 IP vs input' and the 'Dicer-2 IP vs IgG' comparisons as the list of reliable Dicer-2 targets.

2) The authors refer to Fig. 1A for "typical IP efficiency" but it is not possible to determine the efficiency of the IP without any indication of what proportion of the sample was loaded (and the unbound fraction is not shown either).

We thank the reviewer for pointing this out. We have now indicated the percentage of input loaded in all experiments.

3) Figure 1F - What do the error bars represent? Were biological replicates performed? What statistical analysis was done?

We thank the reviewer for this comment. Bar-plots represent the average of three independent biological replicates. Error bars represent standard deviation. Significance was assessed by Student's t-test. We have added this information to the figure legend.

4) The anti-Dicer-2 antibodies generated for this study should be validated - at the least, the authors should provide a full blot with extracts from wild-type and Dicer-2 mutants showing that only a single band corresponding to Dicer-2 is detected by the antibodies.

The anti-Dicer-2 antibodies used in this study have been described in a previous publication from our lab (Coll et al., RNA 2018). As indicated in Methods, they are polyclonal antibodies against the first 151 aa of Dicer-2 that have been affinity-purified in a HiTrap NHS-activated HP column coupled with His-Dicer-2(1-257)-MBP protein. They have indeed been validated by Western-blot against wild type and Dicer-2 null embryos. Partial blots are shown in Figures 2B and 3E. A Figure with full blot is included below for the reviewer (Figure R3).

Figure R3. Characterization of the anti-Dicer-2 antibody. A) Western blot of wt and Dicer-2 null embryo extracts. The asterisk represents a non-specific band. B) The non-specific band is not immunoprecipitated by the anti-Dicer-2 antibody.

5) Please use page numbers.
Page numbers have now been added.

August 8, 2022

Re: Life Science Alliance manuscript #LSA-2022-01417R

Dr. Fatima Gebauer
Centre for Genomic Regulation
Dr Aiguader 88
Barcelona 08003
Spain

Dear Dr. Gebauer,

Thank you for submitting your revised manuscript entitled "Atx2, Tyf and Dcr-2 are components of a non-canonical cytoplasmic polyadenylation complex" to Life Science Alliance. The manuscript has been seen by the original reviewers whose comments are appended below. While the reviewers continue to be overall positive about the work in terms of its suitability for Life Science Alliance, some important issues remain.

Our general policy is that papers are considered through only one revision cycle; however, given that the suggested changes are relatively minor, we are open to one additional short round of revision. Please note that I will expect to make a final decision without additional reviewer input upon re-submission.

Please submit the final revision within one month, along with a letter that includes a point by point response to the remaining reviewer comments. Providing a functional connection to translation, as mentioned by Reviewer 2, is not required.

To upload the revised version of your manuscript, please log in to your account: <https://lsa.msubmit.net/cgi-bin/main.plex>
You will be guided to complete the submission of your revised manuscript and to fill in all necessary information.

B. MANUSCRIPT ORGANIZATION AND FORMATTING:

Sincerely,

Reviewer #1 (Comments to the Authors (Required)):

The revised version of the manuscript by Nadimpalli et al. is significantly improved. However, at the same time, the inherent

difficulty of the system precluded more direct identification of DICER-2 targets and changes in poly(A) tail dynamics. Therefore although the study is worth publishing, it is full of overstatements in its current form, which should be removed. Moreover, in addition to changes in the manuscript text, one aspect still needs to be addressed bioinformatically.

The main issue.

In the section "identification of Dicer-2 mRNA targets," the Authors compared the list of Dicer-2 bound transcripts to potential Wispy polyadenylation substrates identified in a previous study (Cui et al., 2013). The study by Cui et al. used old fashioned error-prone method for poly(A) tail dynamics analysis based on oligo dT enrichment and microarrays. Two more recent studies used TAIL-seq and PAL-seq for this purpose (Genes Dev. 2016 Jul 15;30(14):1671-82; Elife. 2016 Jul 30;5:e16955). I suggest comparing the potential Dicer-2 bound mRNA list with all three datasets.

Major issues (overstatements):

1)

Abstract

"Using RIP-Seq analysis we identify hundreds of novel Dicer-2 target transcripts, ", I would change "novel" into potential. "Our results reveal the composition of a novel cytoplasmic polyadenylation complex that operates during Drosophila early embryogenesis." The composition is not revealed as the complex is not purified to homogeneity. I would change to suggest the existence.

2

Instruction

This process is controlled by the cytoplasmic polyadenylation element binding (CPEB) family of proteins, which bind to U-rich cytoplasmic polyadenylation elements (CPEs) in the 3' UTR of transcripts (Ivshina et al., 2014). The statement is too broad. Cytoplasmic polyadenylation during bone formation seems to be CPEB independent.

3)

The Dicer-2 protein interactome

"Two different bioinformatic procedures were used to identify true Dicer-2 interactors, SAINTexpress and Top3" I suggest changing into " to potentially true
"Altogether, 128 proteins were found to interact with Dicer-2 over the null control, of which 39 were found using both SAINT and Top3. I suggest changing into "Altogether, 128 represented potential interactions with Dicer-2 over, of which 39 were selected using both SAINT and Top3"

4)

Atxin-2 and Twenty-four mediate the interaction of Dicer-2 with Wispy

"Altogether, these results are consistent with a model in which Atx2/Tyf/Wispy form a tight complex that interacts with Dicer-2 (Figure 3F)." There is no data on the stability/tightness of the complex, and the word "tight" should be removed. I would also add information that the exact composition of the complex needs further investigation.

5)

Twenty-four, Atxin-2 and Dicer-2 work in cytoplasmic polyadenylation.

"In sum, our results reveal the composition of a non-canonical cytoplasmic polyadenylation machinery that recruits Wispy for activation of maternal mRNAs, illustrating the growing diversity and plasticity of poly(A) tail length control." The presented results only indicate the presence of complex with such components and suggest such a scenario.

Reviewer #2 (Comments to the Authors (Required)):

The authors have addressed the majority of my concerns. The new data in Figure 3 provide important validation for the protein complex model. It is odd, however, that PABP does not interact with Dicer2 given that they can detect other indirect interactors (e.g., Dicer2 and Tyf). The authors should comment on why PABP is not pulled down in the complex. Also, I do agree with the authors that the "provide clear evidence that Atx2 and Tyf are involved in polyadenylation of Dicer-2 targets, and that these factors interact with the cytoplasmic poly(A) polymerase Wispy" although I still believe that the manuscript would be stronger if there was a functional connection to translation.

Response to reviewers #LSA-2022-01417R**Reviewer #1:**

The revised version of the manuscript by Nadimpalli et al. is significantly improved. However, at the same time, the inherent difficulty of the system precluded more direct identification of DICER-2 targets and changes in poly(A) tail dynamics. Therefore although the study is worth publishing, it is full of overstatements in its current form, which should be removed. Moreover, in addition to changes in the manuscript text, one aspect still needs to be addressed bioinformatically.

The main issue.

In the section "identification of Dicer-2 mRNA targets," the Authors compared the list of Dicer-2 bound transcripts to potential Wispy polyadenylation substrates identified in a previous study (Cui et al., 2013). The study by Cui et al. used old fashioned error-prone method for poly(A) tail dynamics analysis based on oligo dT enrichment and microarrays. Two more recent studies used TAIL-seq and PAL-seq for this purpose (Genes Dev. 2016 Jul 15;30(14):1671-82; Elife. 2016 Jul 30;5:e16955). I suggest used TAIL-seq and PAL-seq for this purpose (Genes Dev. 2016 Jul 15;30(14):1671-82; Elife. 2016 Jul 30;5:e16955). I suggest comparing the potential Dicer-2 bound mRNA list with all three datasets.

We have performed the comparison suggested by the reviewer. The overlap between the three Wispy studies and the list of Dicer-2 targets is now shown in Figure 1E. As the overlap between the three Wispy studies themselves is small, we have considered targets detected in at least one study as our final list of Wispy targets. The number of Wispy targets that are also Dicer-2 targets has increased by 79 with respect to the list in our previous version. Table S1, the main text and the Figure legend have been updated accordingly. We have also updated the gene names in both Tables, as these are constantly updated in databases.

Minor issues (overstatements):

1) Abstract

"Using RIP-Seq analysis we identify hundreds of novel Dicer-2 target transcripts, ", I would change "novel" into potential.

We have changed the wording as suggested by the reviewer.

"Our results reveal the composition of a novel cytoplasmic polyadenylation complex that operates during Drosophila early embryogenesis." The composition is not revealed as the complex is not purified to homogeneity. I would change to suggest the existence.

We have changed the sentence as follows: ' Our results reveal components of a novel cytoplasmic polyadenylation complex that operates during Drosophila early embryogenesis'

2) Introduction

This process is controlled by the cytoplasmic polyadenylation element binding (CPEB) family of proteins, which bind to U-rich cytoplasmic polyadenylation elements (CPEs) in the 3' UTR of transcripts (Ivshina et al., 2014). The statement is too broad. Cytoplasmic polyadenylation during bone formation seems to be CPEB independent.

We have introduced the word 'typically' to acknowledge that CPEB-mediated polyadenylation is the most frequent mechanism, while other mechanisms may exist: ' This process is typically controlled by the cytoplasmic polyadenylation element binding (CPEB) family of proteins...'

3) The Dicer-2 protein interactome

"Two different bioinformatic procedures were used to identify true Dicer-2 interactors, SAINTexpress and Top3" I suggest changing into " to potentially true.

We have changed the wording as suggested by the reviewer.

"Altogether, 128 proteins were found to interact with Dicer-2 over the null control, of which 39 were found using both SAINT and Top3. I suggest changing into "Altogether, 128 represented potential interactions with Dicer-2 over, of which 39 were selected using both SAINT and Top3"

We think the proposed sentence is more confusing. As it appears that the reviewer has a problem with the word 'interact', we have changed the sentence to eliminate this word: "Altogether, 128 proteins were found in the Dicer-2 IP over the null control, of which 39 were found using both SAINT and Top3".

4) Ataxin-2 and Twenty-four mediate the interaction of Dicer-2 with Wispy

"Altogether, these results are consistent with a model in which Atx2/Tyf/Wispy form a tight complex that interacts with Dicer-2 (Figure 3F)." There is no data on the stability/tightness of the complex, and the word

"tight" should be removed. I would also add information that the exact composition of the complex needs further investigation.

The word 'tight' has been removed, and we have added a sentence as suggested by the reviewer.

5) Twenty-four, Ataxin-2 and Dicer-2 work in cytoplasmic polyadenylation.

"In sum, our results reveal the composition of a non-canonical cytoplasmic polyadenylation machinery that recruits Wispy for activation of maternal mRNAs, illustrating the growing diversity and plasticity of poly(A) tail length control." The presented results only indicate the presence of complex with such components and suggest such a scenario.

We do not understand what is the problem with this sentence. Our results do reveal components of a non-canonical cytoplasmic polyadenylation machinery, adding to the growing list of factors involved in poly(A)-tail length control. We have added the word 'assemblies' for clarity.

Reviewer #2:

The authors have addressed the majority of my concerns. The new data in Figure 3 provide important validation for the protein complex model. It is odd, however, that PABP does not interact with Dicer2 given that they can detect other indirect interactors (e.g., Dicer2 and Tyf). The authors should comment on why PABP is not pulled down in the complex.

We have added the following text to the last section of 'Results and Discussion': Surprisingly PABP, an interactor of the Atx2/Tyf complex for translational regulation, was not included in the high-confidence Dicer-2 interactor list (Lim et al., 2011; Lim and Allada, 2013; Zhang et

al., 2013). This is because PABP scored positive only in the Dcr-2 vs IgG IP comparison, suggesting that our strict criteria may exclude true interactors (Table S2).

Also, I do agree with the authors that they "provide clear evidence that Atx2 and Tyf are involved in polyadenylation of Dicer-2 targets, and that these factors interact with the cytoplasmic poly(A) polymerase Wispy" although I still believe that the manuscript would be stronger if there was a functional connection to translation.

We are glad that the reviewer agrees with us.

August 22, 2022

RE: Life Science Alliance Manuscript #LSA-2022-01417RR

Dr. Fatima Gebauer
Centre for Genomic Regulation
Dr Aiguader 88
Barcelona 08003
Spain

Dear Dr. Gebauer,

Thank you for submitting your revised manuscript entitled "Atx2, Tyf and Dcr-2 are components of a non-canonical cytoplasmic polyadenylation complex". We would be happy to publish your paper in Life Science Alliance pending final revisions necessary to meet our formatting guidelines.

Along with points mentioned below, please tend to the following:
-your deposited datasets should now be made publicly available

A. FINAL FILES:

B. MANUSCRIPT ORGANIZATION AND FORMATTING:

Sincerely,

September 6, 2022

RE: Life Science Alliance Manuscript #LSA-2022-01417RRR

Dr. Fatima Gebauer
Centre for Genomic Regulation
Dr Aiguader 88
Barcelona 08003
Spain

Dear Dr. Gebauer,

Thank you for submitting your Research Article entitled "Atx2, Tyf and Dcr-2 are components of a non-canonical cytoplasmic polyadenylation complex". It is a pleasure to let you know that your manuscript is now accepted for publication in Life Science Alliance. Congratulations on this interesting work.

DISTRIBUTION OF MATERIALS:

Again, congratulations on a very nice paper. I hope you found the review process to be constructive and are pleased with how the manuscript was handled editorially. We look forward to future exciting submissions from your lab.

Sincerely,
